# Wound Microbiota and Its Impact on Wound Healing

**DOI:** 10.3390/ijms242417318

**Published:** 2023-12-10

**Authors:** Małgorzata Zielińska, Agnieszka Pawłowska, Anna Orzeł, Luiza Sulej, Katarzyna Muzyka-Placzyńska, Arkadiusz Baran, Dagmara Filipecka-Tyczka, Paulina Pawłowska, Aleksandra Nowińska, Joanna Bogusławska, Anna Scholz

**Affiliations:** 1Ist Department of Obstetrics and Gynecology, Centre of Postgraduate Medical Education, 02-097 Warsaw, Poland; go007@wp.pl (M.Z.); anna.k.orzel@gmail.com (A.O.);; 2Students Research Group of Obstetrics and Gynecology Department at St. Sophia Hospital, 01-004 Warsaw, Poland; pawlowska.agnieszka12@gmail.com (A.P.);; 3Students Scientific Association, Department of Hygiene and Epidemiology, Medical University of Lublin, 20-093 Lublin, Poland; 4Department of Biochemistry and Molecular Biology, Centre of Postgraduate Medical Education, 02-097 Warsaw, Poland; joanna.boguslawska@cmkp.edu.pl

**Keywords:** wounds, microbiota, microbiome composition

## Abstract

Wound healing is a complex process influenced by age, systemic conditions, and local factors. The wound microbiota’s crucial role in this process is gaining recognition. This concise review outlines wound microbiota impacts on healing, emphasizing distinct phases like hemostasis, inflammation, and cell proliferation. Inflammatory responses, orchestrated by growth factors and cytokines, recruit neutrophils and monocytes to eliminate pathogens and debris. Notably, microbiota alterations relate to changes in wound healing dynamics. Commensal bacteria influence immune responses, keratinocyte growth, and blood vessel development. For instance, *Staphylococcus epidermidis* aids keratinocyte progression, while *Staphylococcus aureus* colonization impedes healing. Other bacteria like Group A *Streptococcus* spp. And *Pseudomonas* affect wound healing as well. Clinical applications of microbiota-based wound care are promising, with probiotics and specific bacteria like *Acinetobacter baumannii* aiding tissue repair through molecule secretion. Understanding microbiota influence on wound healing offers therapeutic avenues. Tailored approaches, including probiotics, prebiotics, and antibiotics, can manipulate the microbiota to enhance immune modulation, tissue repair, and inflammation control. Despite progress, critical questions linger. Determining the ideal microbiota composition for optimal wound healing, elucidating precise influence mechanisms, devising effective manipulation strategies, and comprehending the intricate interplay between the microbiota, host, and other factors require further exploration.

## 1. Introduction

Wound healing is a multifaceted and advanced process that can impair various factors. These variables can be categorized as systemic and local variables. The incidence and prevalence of systemic conditions leading to poor wound healing increase with age. Being >60 years is a significant risk factor for impaired healing. Changes in sex hormones also play a role in age-related wound healing deficiencies. Other, better known systemic conditions and variables such as diabetes, obesity, drug administration (e.g., systemic glucocorticosteroids, nonsteroidal anti-inflammatory drugs, chemotherapeutic drugs), alcohol consumption, smoking, inadequate nutrition, and stress, only to mention the most important ones, also impair the biological mechanisms of wound healing [1]. The second group consists of local factors such as oxygenation, infection, foreign bodies, and venous sufficiency. All these factors can subsequently cause an increase in mortality and morbidity, impacting the life of the patients and resulting in significant economic costs. More interesting is the recent increase in evidence that wound microbiota can positively impact the healing process [2]. Commensal bacteria represent a category of microorganisms that coexist symbiotically with the host organism without inducing harm. These bacteria inhabit the body of the host, residing either externally or internally, without precipitating disease or conferring overt advantages. Wound microbiota is defined as a diverse community of microorganisms—bacteria, viruses, fungi and other microbes—which inhabit the wound and influence its physiological function, and also its inflammatory function [3]. Inflammatory responses encompass intricate biological reactions that manifest within the organism as a protective mechanism in response to injury, infection, or other deleterious stimuli. The primary objective of inflammation is the removal of the instigating factor of cellular injury, the elimination of damaged cells and tissues, and the initiation of reparative processes within the tissue [1]. The complexity of the wound healing, interactions between skin compartments, cells, extracellular matrix, and systemic contributions can be divided into four overlapping phases [1]. Several different cells, as well as diverse cytokines, growth factors, and molecules, play a major role in each phase. Because of the intricacy of these processes, possible further research can incorporate cutting-edge techniques and tools such as artificial intelligence (AI) and machine learning (ML) into the analysis, integration of the data, and predictive modelling, as well as the identification of patterns and possible outcomes. The recent improvement of these methods and their predictable advancement can render them indispensable and contribute to the development of scientific research. This review aims to present the current knowledge and to identify possible future research directions to better understand the impact of wound cultures on better wound healing and the leading mechanisms. Focusing on this factor could yield a better therapeutic process, antibiotic administration, and wound care, and possibly new treatment methods.

## 2. Molecular Mechanisms of Wound Healing

Wound healing is a complex, organized process divided into overlapping phases: 1. Hemostasis, 2. Inflammation, 3. Proliferation and 4. Remodeling [1]. The process sequence strictly relies on the interactions between different skin compartments, extracellular matrix (ECM), and systemic contribution. The communication between these variables is facilitated by the expression of adhesive molecules, growth factors, cytokines, and chemokines, such as interleukins (IL-1, IL-6), tumor necrosis factor-alpha (TNFα), platelet-derived growth factor (PDGF), platelet activity factor (PAF), or fibroblast growth factor-2 (FGF-2) [4]. The subsequent stages of wound healing have been presented in Figure 1.

The first phase, hemostasis, begins rapidly after wounding and aims to control bleeding by vasoconstriction and activation of the coagulation cascade. Platelets are activated by exposed subendothelial collagen, and together with thrombin and fibronectin form a clot. The clot controls bleeding and releases factors such as PDGF, tumor growth factor α (TGFα), PAF, fibronectin, and serotonin, thereby initiating inflammation [5]. Immediately after clot formation, cellular signaling results in a neutrophil response. The accumulation of inflammatory mediators subsequently results in vasodilatation and increased vascular permeability, followed by the migration of neutrophils, monocytes, lymphocytes, and fibroblasts [1,6]. All cells are drawn into the injured area by IL-1, TNFα, transforming growth factor β (TGFβ), platelet factor 4 (PF4), and any bacterial factors, and vasodilatation occurs due to the increasing levels of prostaglandins produced by the COX-2 enzyme in endothelial cells [1]. The aforementioned serotonin increases the permeability of vessels and subsequently promotes cell migration. 

Neutrophils are the first cells to be drawn by IL-1, TNFα, TGFβ, and PF4 [2]. Their role is to clear the wounds of invading bacteria and cellular debris. They are also responsible for additional bystander damage caused by the production of proteases and reactive oxygen species (ROS) [4]. The proteases have specific targets. Serine proteases, such as elastase, exhibit wide-ranging specificity, whereas metalloproteinases (MMPs), which contain a zinc ion, specifically target collagen for digestion [1]. The matrix of healthy tissues produces antiprotease proteins to protect itself from further destruction. The second number of cells that increase in the wounded tissue during the inflammatory phase are monocytes that develop into macrophages [2]. This occurs around 48–96 h after injury [1]. Macrophages are essential, as they mediate angiogenesis by producing vascular endothelial growth factor (VEGF), FGF, and TNFα, as well as promoting matrix synthesis regulation via the synthesis of epidermal growth factor (EGF), TGFβ, PDGF, IL-1, and TNFα [5,6]. They are also responsible for further pathogen killing via nitric oxide (NO) generation. TNFα also promotes the expression of MMPs by keratinocytes, fibroblasts, monocytes, and macrophages, which leads to further clearing of the damaged ECM [1]. 

Inflammation is not self-limiting. It must be stopped by “checkpoint controllers of inflammation”, which is a complex process that depends on cell-to-cell interactions and the production of lipoxins (especially lipoxin A4 and lipoxin B5) as the main factor capable of turning off the destructive cycle of inflammation [7]. This happens when platelets and neutrophils adhere to one another [1].

The next phase consists of the proliferation and migration of cells and their subsequent epithelization. The first stimuli for this phase are inflammatory cytokines such as IL-1 and TNFα. They activate *KGF* gene expression in fibroblasts. Conversely, fibroblasts produce and release keratinocyte growth factor (KGF)-1, KGF-2, and IL-6, which induce neighboring keratinocytes to migrate into the wound area, undergo proliferation, and differentiate within the epidermis [1]. They also enhance the transformation of monocytes into macrophages [1]. Epithelial cells begin to reestablish a protective barrier against bacterial invasion and loss of fluid [6]. Keratinocytes, macrophages, fibroblasts, platelets, and other epithelial cells produce VEGF, a chemotactic signal for endothelial cells responsible for forming new capillaries by angiogenesis. These cells are also activated by hypoxia at the beginning of injury. Subsequently, they produce NO, resulting in increased VEGF production. This is only one example of how most processes overlap because of autocrine and paracrine systems. During angiogenesis, fibroblasts proliferate and begin to synthesize type III collagen, glycosaminoglycans, and fibronectin [6]. Triggered by macrophages and secreted TGF β1 and PDGF, fibroblasts develop into myofibroblasts, causing wound contraction [1]. 

The Intact dermis comprises mainly”coll’gen I (approximately 80–90%) and some collagen III (approximately 10–20%). Granulation tissue contains more collagen III, which is relatively low in mature scar tissues. The proportion of these collagens shifts due to the final remodeling process, clinically the most important one [6]. When patients experience issues with matrix deposition, whether due to dietary factors or underlying diseases, it can significantly weaken the strength of the wound. Conversely, excessive collagen synthesis can lead to the formation of hypertrophic scars or keloids [1]. Firstly, the matrix was composed of fibrin and fibronectin. Meanwhile, fibroblasts produce glycosaminoglycans, proteoglycans, and other proteins. This is a preliminary and disorganized matrix framework that is then replaced by collagen. Initially, collagen III was thinner and more permeable than collagen I, enabling cells to migrate through the newly formed matrix [6]. The remodeling of collagen advances over time and is conducted by MMPs, which are influenced by fluctuations in the concentrations of TGF-β, PDGF, IL-1, and EGF [1]. Net collagen production persists for at least 4–5 weeks following injury. The escalated pace of collagen production in wound healing results from an augmented number of fibroblasts and a net rise in collagen output per individual cell [6]. The organization of the collagen in the scar tissue and wound strength will always be less organized than in uninjured skin [1]. 

What is more interesting is that research has shown that wound healing is also impacted by cells both from hair follicles (HF) and intrafollicular epidermis, which migrate to the side of injury [8]. Follicular cells in wound conditions can undergo reprogramming into epidermal progenitors [9], enhancing wound repair. Moreover, the hair cycle and the dynamic equilibrium of bone morphogenetic signaling (BMP) and WNT/ß-catenin signaling within their niche [10]. The latter pathway is pivotal in the regeneration and differentiation of hair follicle stem cells (HFSCs) and is essential for skin reconstruction following wound healing. It stimulates the proliferation of HFSCs and their transformation into dermal papilla cells as well as the contribution of HFSCs to the replenishment of sebaceous glands [11]. This pathway is essential for the restoration of HF and skin. 

In the growth phase of HF, WNT signal expression reaches its peak, activating HFSCs and contributing to wound repair. Conversely, in the resting phase, WNT signal expression diminishes, impacting the healing process [10]. Whyte et al. compared the healing process of wounds in the anagen and telogen phases of hair in mice, proving that the process of re-epithelization in the wound occurred faster during anagen. They concluded that increased WNT signaling in anagen was responsible [10].

Besides WNT/ß-catenin signaling, AKT signaling is another important factor in activating hair follicle stem cells (HFSCs). Following tissue damage, activation occurs in all macrophages, including Ly6C+ inflammatory macrophages and CX3CR1+ tissue-resident macrophages. These cells produce TNF that triggers AKT signaling in HFSC, which leads to telogen–anagen transition and hair follicle regeneration. Wound-induced hair anagen re-entry/growth (WIH-A) takes place in the tissue surrounding the wound site, and the elimination of macrophages, particularly in the initial stages of wound healing, hinders the repair process [12].

Injuries create a chance for microorganisms in the skin’s microbiota and those in the surroundings to infiltrate deeper tissues and locate favorable circumstances for thriving and establishing a colony. Several animal studies have suggested that alterations in the wound microbiota alter the wound healing process [13,14]. Wolcott et al. conducted a survey in which a human chronic wound microbiota, obtained from 50 patients, was inoculated into 47 female damaged mouse models. It produced a similar effect on wound morphology in mice, suggesting the impact of the microbiota on wound healing. Interestingly, the study found that the patient’s entire skin surface tends to be colonized by the microbiota of the chronic wound, resulting in an increased possibility of infection for any future surgery [14]. 

## 3. The Influence of Microbiota on Wound-Healing Mechanisms

The skin microbiota can have both positive and negative effects on wound healing. The variety of commensal bacteria on the skin affects the immune response within the wound, is crucial in the maintenance of the epithelial barrier function, and may protect against or limit bacterial wound infections. Several mechanisms link the skin microbiota to wound healing, which differ depending on the type of bacteria [3]. The skin microbiota influences various processes in the skin, such as keratinocyte proliferation, epithelial differentiation, epidermal blood vessel growth, and cell signaling [14,15]. 

One of the most common commensal *Staphylococcus epidermidis* upregulates Toll-like receptors (TLR) and downstream modulation of TNF-α, which, through skin CD8+ T cells, accelerates the progression of keratinocytes. *S. epidermidis* produces lipoteichoic acid, which reduces inflammation via TLR2 signaling [3]. 

Another mechanism is associated with AMP expression in keratinocytes, such as small cationic beta-defensin molecules (hBD), which play a role in epithelial differentiation. Their differential expression amplifies the innate immune response to bacteria. This mechanism is triggered by both *S. epidermidis* and the typically low abundance of *Staphylococcus aureus* [3]. 

*S. aureus* is a normal commensal of the skin microbiota. However, it produces superantigens (Sag), which are harmful in high systemic concentrations but beneficial in small amounts. Sag production, in comparison with non-productive strains of *S. aureus*, results in decreased skin inflammation and purulence in wounds because of the declining local production of IL-17 and neutrophil chemotactic factors. However, *S. aureus* colonization may negatively affect wound healing because of elevated levels of keratinocyte cytokines and chemokine ligands, IL-1B, IL-6, CXCL-1, and TNF-α [3]. 

Group A *Streptococcus* spp. Is usually pathogenic. However, it can activate plasminogen, which is responsible for keratinocyte chemotaxis and potential re-epithelization of wounds. In addition, *S. epidermidis* stimulates the production of AMPs [3]. 

Colonization of host epithelial tissue by *Pseudomonas* spp. Has different effects depending on the absence or presence of infection. Low levels increase the rate of blood vessel growth through keratinocyte growth factor-1 and accelerate epithelialization. Additionally, *Pseudomonas* spp. Stimulates the TAK1/MKK/p38 signaling pathway, which induces cell apoptosis and inhibits tissue regeneration in the presence of infection; however, in the absence of infection, it represses apoptosis and improves wound healing [16,17].

Another mechanism is represented by *Corynebacterium jeikeium*, which protects the host epidermis from free radical oxygen species via the production of superoxide dismutase and manganese acquisition. *Propionibacteria* induce the expression of TLR2 and TLR4 in keratinocytes and produce bacteriocins that protect the sebaceous ducts from other pathogenic bacteria [16,17].

Ernlunf et al. investigated the bacterial microbiota of 11 patients, before and after washing, with venous stasis ulcers before and after wound cleaning. They showed that washing venous stasis ulcers did not change the diversity of the ulcer microbiota [18]. Plichta et al. studied the microbiota of wounds after burn injuries. They showed that cutaneous burns significantly changed the skin microbiota. In these types of wounds, there was an increased abundance of thermophilic bacteria, such as *Aeribacillus* spp., *Caldalkalibacillus* spp., and *Nesterenkonia* spp. In treating burns, wounds have proven to be beneficial topical antibiotics against *Pseudomonas* spp. Additionally, microbiota disorders are associated with complications, such as burn infection correlated with *Corynebacterium* spp. [19]. Liu et al. also investigated the correlation between changes in the skin microbiota and burn wound healing in 6 patients with burn wounds and 13 patients with healthy skin; however, despite the increased abundance of *Firmicutes* and *Staphylococcus* spp., all of these wounds healed successfully, and the impact of changing the microbiota on the rate of wound healing has not been studied [20].

Wounds after blunt or penetrating trauma may also alter the skin microbiota. *Staphylococcus*, *Corynebacterium*, *Streptococcus*, *Acinetobacter*, *Anaerococcus*, *Finegoldia*, and *Pseudomonas* were the dominant bacteria [15]. 

Tuttle et al. showed that wounds that had not healed for six months were characterized by higher bacterial abundance and diversity [21]. Moreover, the diversity of the bacteria in the wound depended on the type of the wound. An observational study involving 50 specimens from chronic wounds in adult patients indicated a higher occurrence of biofilm formation in 60% of chronic wounds, while only 6% of acute wounds displayed such a phenomenon. Chronic wounds exhibit an extended inflammatory phase, and it appears that microorganisms play a role in perpetuating this inflammatory response. Consequently, bacteria are afforded an extended duration during which they can access nutrients derived from the host [22]. It was demonstrated that *Streptococcus* spp. Appeared the most commonly in chronic wounds, with *S. aureus* and *S. epidermidis* as the predominant species and more pronounced presence of *P. aeruginosa*, indicating a greater overall abundance and showcasing the formation of biofilms.

In summary, the skin microbiota affects healing primarily by stimulating cell signaling pathways, the effect of which depends on the high diversity of bacterial strains present on the host skin and the presence of infection. These effects are summarized in Table 1. However, because of the less defined role of cutaneous microbiota, and controversial and contradictory findings from existing pre-clinical and clinical studies, additional investigation is required to delineate the function of the human skin microbiota in acute and chronic wound recovery.

## 4. Antibiotic Prophylaxis in Various Specialties

Surgical site infections are a major problem in the surgical fields of medicine, such as surgery and gynecology. Antibiotic prophylaxis is used to reduce the risk of surgical site infections (SSIs). Administration of antibiotics during cesarean section has been shown to reduce SSIs, but it is controversial whether antibiotics should be administered before skin incision or after cord clamping. The type and duration of optimal preoperative antibiotics during cesarean section are yet to be determined [23]. 

In their meta-analysis, Liu et al. provided compelling evidence on the efficacy of prophylactic intravenous antibiotics in surgical settings. Their findings indicate a significant reduction in the risk of surgical site infections (SSIs) following breast cancer surgery when prophylactic antibiotics are administered preoperatively, as opposed to a placebo. This conclusion is substantiated by high-certainty evidence from six trials, encompassing a total of 1708 participants. Moreover, the study highlights the effectiveness of antibiotic prophylaxis in cesarean sections. The preemptive use of antibiotics in these procedures, compared to their non-use, is associated with a likely reduction in SSI risk. This is supported by moderate-certainty evidence from an extensive dataset of 82 trials involving 14,407 participants. In the realm of hernia repair surgeries, the analysis similarly suggests that the prophylactic administration of antibiotics can lead to a decrease in SSI risk when compared with placebo or no antibiotic treatment. This inference is drawn from moderate-certainty evidence across 17 trials, which included 7843 participants [24]. 

Futier et al.’s study, a multicenter, randomized, double-blind, placebo-controlled trial, demonstrated that a combined regimen of oral and intravenous antibiotic prophylaxis is more effective in reducing the risk of surgical site infections (SSIs) than either oral or intravenous antibiotic administration alone [25]. 

Antibiotics targeting specific bacteria can also be used in preoperative prophylaxis. One example is the use of intranasal mupirocin in MRSA or MSSA carriers before elective total joint arthroplasty. Sporer et al. showed a 69% reduction in the prevalence of SSIs in patients undergoing total joint arthroplasty with *S. aureus* decolonization [26].

Perioperative antibiotic prophylaxis is also used in plastic surgery, for example, during breast reconstruction surgery when there is a risk of implant infection. Liu et al. conducted a significant study in this context, where they meticulously evaluated 507 consecutive cases involving expander/implant-based breast reconstruction, ensuring a minimum follow-up period of 1 year for each case. Their findings revealed a pivotal insight: a one-week postoperative antibiotic regimen was correlated with a higher incidence of surgical site infections (SSIs) when compared to longer-duration antibiotic courses [27]. 

In the field of neurosurgery, the incidence of surgical site infections (SSIs) is relatively low, but it varies notably across different clinical scenarios and studies. For instance, in the context of cervical spine surgery, SSI rates have been reported to range from 0.7% to 11.9%. This variation is reflective of the diverse clinical settings and methodologies employed in different studies. The occurrence of SSIs in neurosurgical procedures, including cervical spine surgeries, is influenced by multiple factors such as the nature of the surgical procedure, patient-specific characteristics, and the implementation of preventive strategies. A noteworthy study that analyzed a cohort of 797 patients undergoing cervical spine surgery reported an even lower incidence of SSIs (0.25%) [28]. In their retrospective analysis, Cao et al. scrutinized a significant sample of 808 cases involving patients who underwent clean neurosurgical operations. The study revealed that antibiotic prophylaxis did not significantly prevent postoperative infections during clean neurosurgical procedures. Intriguingly, the analysis identified two independent risk factors for postoperative infections: cerebrospinal fluid leakage and the duration of the surgical procedure [29]. 

Surgical site infections are a severe complication of cardiac surgery. Perioperative antibiotic prophylaxis is highly recommended in this type of surgery. Mertz et al. showed in a systematic review and meta-analysis that a longer term (>24 h) was associated with reduced sternal SSIs compared to short-term antibiotic prophylaxis [30]. 

Mülholfer et al. executed a comprehensive study focusing on skin culture analyses in patients scheduled for elective hip arthroplasty. This study encompassed a diverse group of 90 patients, which included 63 individuals undergoing primary total hip arthroplasty (THA) and 27 patients undergoing revision THA. A significant finding from this study was that 34.7% of the bacterial strains identified were resistant to cloxacillin, a common antibiotic. However, notably, none of these strains exhibited resistance to vancomycin. These findings suggest vancomycin as a potentially more effective prophylactic antibiotic, especially in the context of the observed resistance patterns [31,32]. Likewise, Bosco et al. conducted a significant study focusing on the skin microbiota of patients undergoing elective knee or hip arthroplasty, involving a substantial cohort of 10,084 patients. The research aimed to assess the effectiveness of enhanced antibiotic prophylaxis. The results were noteworthy: when gentamicin or aztreonam was incorporated into the antibiotic prophylaxis regimen, there was a substantial reduction in the incidence of surgical site infections (SSIs). The reported SSI rate was 0.55%, which marked a significant decrease compared to historical control data [31,33]. 

## 5. Clinical Application

Understanding the impact of the microbiota on wound healing could lead to new healing pathways, for instance by modulating the immune response, promoting tissue repair, or modulating inflammation. These pathways may provide new targets for therapeutic interventions that promote wound healing.

Recent research has unveiled intriguing insights into the impact of probiotic bacteria on collagen production. Notably, in vivo studies have identified specific bacterial strains, such as *Staphylococcus epidermidis*, that influence collagen synthesis. These bacteria appear to ferment a distinct compound, termed CIN, yielding metabolites that activate the FfaR2 receptor. This receptor engagement initiates a signaling cascade involving p-ERK, culminating in the upregulation of type I collagen production [34]. Similarly, another bacterium, *Lactobacillus plantarum*-GMNL6, has been observed to enhance type I collagen synthesis in vivo [35].

Empirical evidence supporting these findings has primarily emerged from animal models, particularly studies conducted on rats, which have demonstrated a marked increase in tissue type I collagen following probiotic administration [36]. However, investigations extending these findings to human subjects remain relatively scarce. A noteworthy exception is a recent clinical study which administered a 12-week probiotic supplementation regimen, comprising *Lactobacillus acidophilus*, *Lactobacillus casei*, *Lactobacillus fermentum*, and *Bifidobacterium bifidum* (each at 2 × 10^9^ CFU/g), administered to diabetic patients afflicted with diabetic foot ulcers [37].

The clinical application of treatments targeting microbiome changes has already been exploited in patients with atopic dermatitis. Recent investigations have revealed a deficiency in the skin of AD patients with antimicrobial peptides (AMPs), specifically β-defensins-2 and -3 [38,39]. This deficiency compromises the ability of the skin to combat infections caused by pathogens, such as *Staphylococcus aureus* [40]. Notably, studies have demonstrated that treatment with a lotion containing *Lactobacillus johnsonii*, a prevalent probiotic strain, significantly improved clinical outcomes and reduced *Staphylococcaceae* bacterial colonization in AD patients [41]. Gueniche et al. found that the topical application of *Vitreoscilla filiformis* improved the healing of skin lesions in patients with AD. This may have been partly due to the reduction of *S. aureus* and its direct immunomodulatory effect [42]. 

Furthermore, the study revealed that a particular strain of bacteria, *Staphylococcus epidermidis*, could prevent skin infections by producing antimicrobial peptides. *S. epidermidis* produces antibacterial peptides known as lantibiotics or bacteriocins, such as epidermin, epilancin K7, epilancin 15×, Pep5, and Staphylococcin 1580. These peptides have antimicrobial properties and can inhibit the growth of other bacteria, including pathogens, such as *Staphylococcus aureus* and *Streptococcus pyogenes*. In addition to these direct mechanisms, *S. epidermidis* establishes a mutualistic relationship with the host, particularly the skin. It can interact with the host immune system through Toll-like receptor (TLR) signaling, influencing the innate immune response of keratinocytes. This interaction “primes” the immune system of the skin, enabling a more effective defense against harmful pathogens. This suggests that probiotics containing this bacterial strain could be used as a prophylactic approach to prevent wound infections. Moreover, this study found that removing *S. epidermidis* through excessive use of topical antibiotics could harm the host for two reasons. First, it eliminates the antimicrobial peptides of bacteria, allowing potentially harmful organisms to colonize the skin more easily. Second, without *S. epidermidis*, the skin may be less effective in preventing infections. This suggests that a more targeted approach to antimicrobial therapy may be more effective in promoting wound healing [43]. 

Moreover, recent research has elucidated the significant role of the microbiome in facilitating tissue repair, a pivotal process in the healing of wounds. Empirical data from animal-based studies underscore the efficacy of probiotics in diminishing bacterial load and augmenting tissue regeneration. A notable investigation highlights the capacity of *Lactobacillus plantarum* to impede acyl-homoserine lactone (AHL) synthesis in *Pseudomonas aeruginosa*. This interference disrupts the quorum-sensing mechanism of the bacterium, culminating in a diminished production of critical virulence factors, such as elastase and biofilm formation. Consequently, this mitigates the virulence of *P. aeruginosa*, rendering it more vulnerable to the host’s immune defenses and therapeutic interventions. The primary contribution of *Lactobacillus plantarum* in tissue repair lies in its ability to inhibit the virulence and proliferation of *Pseudomonas aeruginosa*, obstruct its biofilm development, and bolster the host’s immune response [44]. Additionally, research has demonstrated that extracts from *Lactobacillus reuteri* can potentiate the capabilities of gingival mesenchymal stem cells (GMSCs), thereby expediting wound healing. This finding provides valuable insights into the mechanisms through which probiotics facilitate tissue repair, especially in the realms of oral health and wound recovery [45].

Modulation of inflammation is an essential pathway for wound healing. Inflammation is necessary to remove damaged tissues and initiate recovery; however, excessive inflammation can delay healing. Certain bacteria in the skin microbiome can activate immune cells called gamma delta T cells, which play crucial roles in wound healing. Modulation of the skin microbiome to promote the growth of these beneficial bacteria could be a new pathway for enhancing the immune response and promoting wound healing [46]. Additionally, inflammation is necessary to remove damaged tissue and initiate recovery; however, excessive inflammation can delay healing. Manipulating the gut microbiome to promote the growth of beneficial bacteria that modulate inflammation could be a novel pathway for promoting wound healing [47]. There are also studies indicating that some bacteria, by regulating the inflammatory response, contribute to healing through an increase in IL-10 One study shows a direct clinical link to bacterial strains *Lactobacillus bulgaricus* and *Lactobacillus plantarum*. The treatment with *Lactobacillus bulgaricus* and *Lactobacillus plantarum* was found to accelerate the healing process of diabetic wounds. It modulated the inflammatory cells in the wound sites and altered the mRNA levels of inflammatory cytokines. These probiotics can improve the healing of diabetic wounds by regulating inflammation [48].

Additional therapeutic strategies rooted in the microbiome may encompass the utilization of phage-targeted interventions [49]. Bacteriophages, also known as phages, are viruses with the unique ability to infect and replicate within bacterial cells, while abstaining from replication within eukaryotic cells, thus rendering them amenable to human applications. Notably, phages are acknowledged as the most abundant naturally occurring biological entities on earth, boasting a rich history of deployment in the treatment of bacterial infections in human subjects. Furthermore, phages exhibit an extraordinary degree of specificity, often targeting bacteria at the species and, in some cases, strain levels, endowing them with the capability to selectively infect and combat bacterial populations. The application of phages has yielded successful outcomes in the management of *Staphylococcus aureus* infections and the reduction of *S. aureus* biofilm biomass [50]. Moreover, topical phage therapies have proven effective in addressing a diverse array of chronic, refractory cutaneous wounds, including those associated with venous stasis, burn injuries, and diabetic ulcers [51].

Lastly, another study found that the skin microbiome varied between individuals and was influenced by age, sex, and ethnicity. This suggests that a personalized approach to wound care, considering the patient’s unique microbiome, could be more effective in promoting wound healing [52]. Our understanding of the intricate interactions within the microbiome and their impact on human health is still evolving. Ensuring the safety and efficacy of microbiome-based therapies is critical, given their potential to alter human microbiota.

## 6. Conclusions

Wound healing is a complex process that influences various systemic and local factors. Age, sex hormones, diabetes, obesity, drug administration, alcohol consumption, smoking, inadequate nutrition, stress, and other systemic conditions can impair wound healing, as well as local factors such as oxygenation, infection, foreign bodies, and venous sufficiency. The skin microbiome has both positive and negative effects on wound healing, and the diverse range of commensal bacteria on the skin influences the immune response within the wound, which may either protect against or limit bacterial wound infection. Understanding the impact of wound culture and the molecular mechanisms of wound healing are crucial for developing new treatments that can accelerate the process. Antibiotic prophylaxis is an essential preventive measure in various surgical specialties to reduce the risk of SSIs. Preoperative antibiotics effectively reduced the risk of SSIs after hernia repair, cesarean section, and breast cancer surgery. In contrast, antibiotics targeting specific bacteria can be used before elective total joint arthroplasty. Perioperative antibiotic prophylaxis is also used in plastic surgery, neurosurgery, and cardiac surgery; however, its effectiveness varies depending on the type and duration of antibiotic administration.

Enhanced comprehension of the impact of the microbiome on wound healing has opened new possibilities for wound care strategies and therapeutic interventions. Manipulating the microbiome can promote wound healing through various pathways such as modulating the immune response, promoting tissue repair, and modulating inflammation. For example, certain bacteria in the skin microbiome can activate immune cells that play a critical role in wound healing. In contrast, specific bacterial strains can promote tissue repair by secreting molecules that aid in the healing process. Researchers can develop targeted therapeutic interventions to promote wound healing by identifying and manipulating the beneficial bacteria in the microbiome. 

However, challenges exist in establishing causation versus mere correlation in the microbiome’s impact on wound healing. The complexity of the microbiome, along with numerous confounding factors such as genetic predispositions, environmental influences, and the presence of other health conditions, complicates the determination of a direct causal relationships. These factors can significantly affect microbiome dynamics, potentially affecting the interpretation of research findings.

Despite remarkable strides in understanding the microbiome’s role in wound healing, key questions persist in this evolving field. A primary area of investigation is identifying the ideal microbiome composition that fosters optimal wound healing. While it is established that certain bacterial types can either promote or hinder healing, pinpointing the precise bacterial balance conducive to the best healing outcomes remains an intricate challenge demanding further research [2]. Moreover, the specific bacteria inhabiting a wound are critical. Each bacterial species uniquely influences the healing trajectory, highlighting the need for detailed research focused on identifying and comprehending the wound microbiome. Distinguishing between beneficial and harmful bacteria is crucial for developing effective treatment strategies tailored to the microbial environment of individual wounds. Additionally, a comprehensive understanding of the mechanisms by which the microbiome affects wound healing is essential. Early research indicates the microbiome’s roles in modulating inflammation, promoting collagen increase, and generating antimicrobial substances. However, these studies only scratch the surface, and a more in-depth exploration is necessary to unravel the full extent of these biological processes [15,53]. Furthermore, research is needed to ascertain the most effective methods for microbiome manipulation to enhance wound healing. The potential of probiotics, prebiotics, and antibiotics in this context shows promise, but more work is required to determine the best practices for using these tools to improve healing outcomes [15]. Finally, the microbiome ecosystem is a complex network, interacting not only with the host but also with various other factors that influence wound healing [54,55]. Understanding these interactions in depth is crucial for leveraging them to optimize wound recovery. This necessitates more extensive research to untangle the intricate web of relationships within the microbiome ecosystem and harness its full potential in wound healing applications.

## Figures and Tables

**Figure 1 ijms-24-17318-f001:**
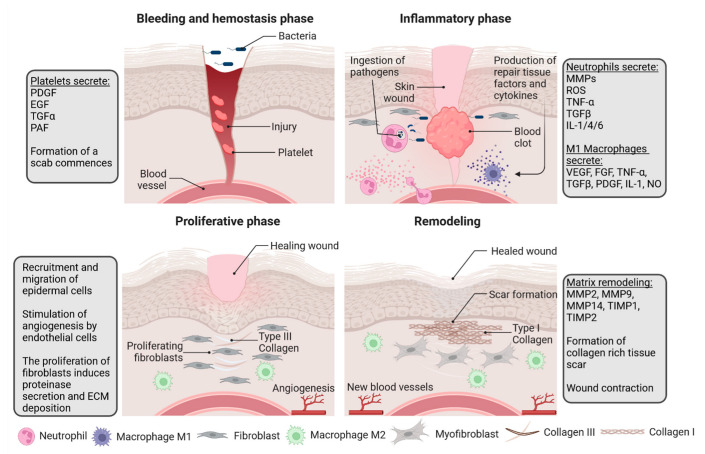
Skin wound healing stages. Bleeding and hemostasis: shortly after bleeding, arteries contract to halt bleeding after vascular damage. Inflammation: a clot forms one to three days after damage. Wounds have drawn inflammatory cells. Neutrophils produce ROS, NO, AMPs, TNFα, IL-1B, IL-6, CXCL2/8, and MCP-1. Proliferative phase: macrophages clean dead tissue and debris. They release IL-1, TNFα, PDGF, VEGF, and TGF-β1. In the wound bed, blood vessels grow. Wound-activated fibroblasts deposit collagen. Remodeling: wound contraction, collagen III replacement by collagen I, and proteases and other enzymes rebuild the extracellular matrix. Created with BioRender.com (accessed on 1 October 2023).

**Table 1 ijms-24-17318-t001:** Type of bacteria and its mechanism affecting wound healing [16,17].

Bacteria	Mechanism
*S. epidermidis*	Upregulation of TLR, downstream modulation of TNF-α
*S. aureus*, *S. epidermidis*	AMPs expression on keratynocytes
Group A *Streptococcus*	Activation of plasminogen
*Pseudomonas*-absence of infection	Stimulation of angiogenesis through of keratinocyte growth factor-1
*Pseudomonas*-presence of infection	Inducing cell apoptosis through TAK1/MKK/p38 signaling pathway
*Corynebacterium jeikeiu*m	production of superoxide dismutase and manganese acquisition
*Propionibacteria*	inducing expression of TLR2 and TLR4

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
