# Peer review of "Wound Microbiota and Its Impact on Wound Healing"

_ijms, 2023, doi:10.3390/ijms242417318_

Round 1
Reviewer 1 Report
Comments and Suggestions for Authors
The article titled "Wound Microbiota and Its Impact on Wound Healing" by Małgorzata Zielińska and others is a review that primarily explores the impact of wound microbiota on wound healing, emphasizing the crucial role of microbiota in the wound healing process. The article discusses in detail the various stages of wound healing, including hemostasis, inflammation, and cell proliferation, and explains how the microbiota influences these processes. It points out that symbiotic bacteria such as Staphylococcus epidermidis promote the growth of keratinocytes, while the colonization of Staphylococcus aureus inhibits healing. Additionally, the article discusses the prospects of microbiota-based wound care in clinical applications, including the use of probiotics and specific bacteria like Acinetobacter baumannii to aid tissue repair.
This review is comprehensive, but considering that the International Journal of Molecular Sciences (IJMS) has very high standards for reviews, I believe it has some flaws. I would like suggestions for improvement.
1.The Influence on Wound Healing-Mechanisms Part I think we can add these. Please don't copy directly.
While the article provides an overview of the relationship between the microbiome and wound healing, it may lack deeper mechanistic explanations in certain aspects. For instance, the mechanisms of Skin wound healing are now quite clear. Research has shown that the alternating active and dormant phases of HFSCs are regulated by the dynamic equilibrium of BMP and WNT signaling within their niche (Chua, A. W., J Invest Dermatol 2011), (Whyte, J. L., PLoS One 2013) (Hsu, Y.C., Nat Med 2014). Besides WNT signals, AKT is another key player in triggering β-catenin signal activation within hair follicle stem cells (HFSCs), with macrophages significantly contributing to this mechanism. When an injury occurs, hair follicle keratinocytes generate the chemokine CCL2, which attracts macrophages to the site (Kobayashi, T., 2019). During the hair follicle's growth phase, there's an increase in macrophage numbers, particularly CX3CR1 bone-marrow-derived macrophages (Wang, X., Nat Commun 2017) (Han, J., iScience 2023) (Chen, H, Theranostics 2019), which release TNF and TGFβ1 (Rahmani, W., J Invest Dermatol 2018). TNF, through the AKT/β-catenin signaling axis, stimulates HFSCs, leading to wound-induced hair anagen re-entry/growth (WIHA) and wound-induced hair follicle neogenesis (WIHN). The TGFβ1 signal plays a critical role in WIHA/WIHN and may initiate hair follicle regeneration via the AKT/PI3K pathway.
2.This should be added to the Conclusion part or mechanism part. Please don't copy directly. You can fill out the Mechanism part in more detail.
The WNT/β-catenin signaling plays a crucial role in the regeneration and differentiation of hair follicle stem cells (HFSCs), and is vital for skin reconstruction after wound healing. Its primary function is to stimulate the proliferation of HFSCs and their transformation into dermal papilla (DP) cells (Rahmani, W. Dev Cell 2014) (Abbasi, S.; Cell Stem Cell 2020) (Han, J. Front Cell Dev Biol 2022). During the process of wound repair, this pathway is key to the restoration of hair follicles and skin. Differentiating fibroblasts take on a more prominent role in the repair of damaged skin. These cells not only combine with new hair follicles to form true DP/DS cells supporting continuous hair follicle regeneration but also participate in the reconstruction of interstitial spaces and aid in the regeneration of the dermis. Furthermore, the WNT/β-catenin pathway promotes the accumulation of HFSCs and the filling of sebaceous glands (SGs). HFSCs and SG cells filled within these glands can further develop into sebaceous cells, playing a positive role in wound healing (Han, J., iScience 2023) (Veniaminova, N., Cell Rep 2023). Additionally, the hair growth cycle impacts wound repair. During the growth phase, WNT signal expression is at its peak, with HFSCs being active and aiding in wound repair. In contrast, during the resting phase, WNT signal expression decreases, affecting the healing process.
I'm sure there will be more than just this literature, It would be better if it could be better added to the mechanism section.
3.The article discusses the potential impact of microbiota on wound healing, but its actual clinical application may still be immature. More clinical studies are needed to validate these findings, and the article should focus on presenting relevant clinical evidence and discuss treatment methods based on these findings in the review.
4. Can slightly enrich the unresolved questions and challenges in the relationship between the microbiome and wound healing in conclusion.
Comments on the Quality of English LanguageMinor editing of English language required.
Author Response
Dear Reviewer,
thank you for your response and the comments regarding the work.
- The section regarding the Wound Healing-Mechanisms Part was enriched according to your suggestions
- The deepened research was conducted regarding the informations you provided and we added it to the conclusion part
- We enclosed more clinical aspects in the work
- More precise evaluation of relationship between the microbiome and wound healing was performed in conclusions
Thank you so much for your time and help.
Best regards,
Anna Orzeł
Reviewer 2 Report
Comments and Suggestions for Authors
This paper explores the intricate relationship between wound microbiota and the phases of wound healing, emphasizing the role of commensal bacteria in modulating immune responses, keratinocyte growth, and blood vessel development. While presenting promising clinical applications, the review underscores the need for further research to determine the ideal microbiome composition, elucidate precise influence mechanisms, and devise effective manipulation strategies to enhance immune modulation and tissue repair. Authors had presented the topic well, but there are some major areas which needs significant improvement to make the paper novel as per IJMS high quality publication as suggested blow in my comments. Addressing these points will contribute to the overall clarity, depth, and impact of the paper on wound microbiota and its influence on wound healing.
Clarify the introduction to provide a more detailed background on wound healing processes. Consider incorporating recent key findings and explaining the increasing relevance of microbiota in wound healing citing some recent reports on page 2 https://doi.org/10.1016/j.crtox.2023.100118 with the sentence ´identify possible future research directions to better understand the impact of wound cultures´ which presented growing therapeutics diversity in wound medicine supported with the AI & ML advances. This will engage readers and highlight the significance of the paper.
Expand the literature review section to include a more comprehensive survey of recent studies on wound microbiota. Discuss various methodologies used in microbiome analysis and highlight any inconsistencies or gaps in current knowledge that the paper aims to address.
Clearly define key terms such as "microbiome," "commensal bacteria," and "inflammatory responses" early in the paper. Providing concise definitions will aid readers, especially those less familiar with microbiology, in understanding the subsequent content.
Elaborate on the methodologies employed in studies cited. Include information on sample size, patient characteristics, and any relevant controls. This will enhance the paper's credibility and help readers evaluate the validity of the presented findings.
Expand the section discussing the mechanisms through which microbiota influences wound healing. Provide detailed insights into how specific bacteria influence immune responses, keratinocyte growth, and blood vessel development. Use molecular and cellular details to strengthen the argument.
If available, include quantitative data to support statements about the impact of specific bacteria on wound healing. This can include statistical analyses, graphical representations, or tables that provide a clearer picture of the observed effects.
Strengthen the clinical applications section by providing more concrete examples of ongoing or completed clinical trials related to microbiome-based wound care. Discuss challenges faced in translating research findings to clinical settings and propose potential solutions.
Provide more detailed information on the mechanisms through which specific bacteria, such as Acinetobacter baumannii, aid tissue repair. Include information on the molecules secreted and their roles in promoting wound healing. In addition, cite a recent report https://doi.org/10.1080/10496475.2023.2267467 with the sentence ´Additional therapeutic strategies rooted in the microbiome may encompass the utilization of phage-targeted interventions´.
Explicitly discuss the limitations of current microbiome research in the context of wound healing. Address any challenges in establishing causation versus correlation and acknowledge potential confounding factors that might affect microbiome dynamics.
Expand the section on future research directions to guide researchers in the field. Discuss potential experiments, technologies, or methodologies that could further elucidate the microbiome's role in wound healing and address the lingering critical questions mentioned in the paper.
Ensure consistency in the usage of terminology throughout the paper. For example, consider using either "microbiota" or "microbiome" consistently to avoid confusion.
Strengthen the concluding remarks by summarizing the key findings and their implications for future research and clinical applications. Emphasize the significance of understanding the microbiome in wound healing and highlight the potential impact on personalized therapeutic approaches.
Comments on the Quality of English LanguageMinor editing of English language required
Author Response
Dear Reviewer,
thank you so much for your time and presented suggestions. The enclosed manuscript was corrected according to your marks:
- The introduction on wound healing processes was enriched with the work proposed by you.
- We clearly presented methodologies used in microbiome analysis and highlighted possible gaps in current knowledge
- Proposed definitions were presented in the work
- The informations regarding methodologies employed in studies cited and molecular and cellular mechanisms of wound healing were presented
- We provided the informations based on clinical trials according to your suggestions
- The impact on wound healing on specific bacteria was presented in the work
- We expanded the limitations of current microbiome research and future possibilities for the researchers in subject of microbiome and wound healing
- We corrected the conclusions according to your suggestions
Thank you so much for your time and complex revision.
Looking forward to hearing from you.
Best regards,
Anna Orzeł
Round 2
Reviewer 1 Report
Comments and Suggestions for Authors
I think this manuscript will become a very good manuscript after correction
Reviewer 2 Report
Comments and Suggestions for Authors
accept